# Rapid Bacterial Recognition over a Wide pH Range by Boronic Acid-Based Ditopic Dendrimer Probes for Gram-Positive Bacteria

**DOI:** 10.3390/molecules27010256

**Published:** 2021-12-31

**Authors:** Ayame Mikagi, Koichi Manita, Asuka Yoyasu, Yuji Tsuchido, Nobuyuki Kanzawa, Takeshi Hashimoto, Takashi Hayashita

**Affiliations:** 1Department of Materials and Life Sciences, Faculty of Science and Technology, Sophia University, 7-1 Kioi-cho, Chiyoda-ku, Tokyo 102-8554, Japan; iris.14mls@eagle.sophia.ac.jp (A.M.); k-manita-hfi@eagle.sophia.ac.jp (K.M.); a-yoya-3nec@eagle.sophia.ac.jp (A.Y.); y-tsuchido@aoni.waseda.jp (Y.T.); n-kanza@sophia.ac.jp (N.K.); t-hasimo@sophia.ac.jp (T.H.); 2Department of Life Science and Medical Bioscience, School of Advanced Science and Engineering, Waseda University (TWIns), 2-2 Wakamatsu-cho, Shinjuku-ku, Tokyo 162-8480, Japan

**Keywords:** bacterial recognition, dendrimer, dipicolylamine, *E. coli*, phenylboronic acid, *S. aureus*

## Abstract

We have developed a convenient and selective method for the detection of Gram-positive bacteria using a ditopic poly(amidoamine) (PAMAM) dendrimer probe. The dendrimer that was modified with dipicolylamine (dpa) and phenylboronic acid groups showed selectivity toward *Staphylococcus aureus*. The ditopic dendrimer system had higher sensitivity and better pH tolerance than the monotopic PAMAM dendrimer probe. We also investigated the mechanisms of various ditopic PAMAM dendrimer probes and found that the selectivity toward Gram-positive bacteria was dependent on a variety of interactions. Supramolecular interactions, such as electrostatic interaction and hydrophobic interaction, per se, did not contribute to the bacterial recognition ability, nor did they improve the selectivity of the ditopic dendrimer system. In contrast, the ditopic PAMAM dendrimer probe that had a phosphate-sensing dpa group and formed a chelate with metal ions showed improved selectivity toward *S. aureus*. The results suggested that the targeted ditopic PAMAM dendrimer probe showed selectivity toward Gram-positive bacteria. This study is expected to contribute to the elucidation of the interaction between synthetic molecules and bacterial surface. Moreover, our novel method showed potential for the rapid and species-specific recognition of various bacteria.

## 1. Introduction

Multidrug-resistant bacteria, the development of which has been accelerated by the excessive use of antibiotics, has become a global issue in the past decade [1,2]. As the abuse of antibiotics stimulates undesired genetic mutations in bacteria, specific antibiotic usage is important from the standpoint of achieving sustainable development goals [3,4]. The need for a bacterial recognition method that would enable the determination of the specific dose of an antibiotic is apparent because existing bacterial recognition methods require several days for bacterial culture [5] or expensive reagents [6,7]. 

This problem has attracted the interest of many research groups, and novel bacterial recognition methods [8,9] or microbial agents [10] have been studied. Research has focused on ways to reduce testing time, improve protocol design, or increase sensitivity. For instance, analytical methods that use matrix-assisted laser desorption ionization time-of-flight mass spectrometry [11] to shorten the testing time have been reported. Other research groups have conducted biological analyses, such as an immunosensing [12], enzyme-linked immunosorbent assay [13], or the polymerase chain reaction [14,15], which have high sensitivity and are species-specific.

As is the case with biologists, chemists have taken up the challenge of developing nanoprobes that can recognize bacterial structure through supramolecular interaction or binding. Mannose-modified nanoprobes were able to recognize some bacterial species [16,17]. Silver nanoparticles with a cell wall binding domain were also able to recognize certain bacteria [18]. Some researchers have designed nanoprobes with the bis(2-pyridylmethyl)amine group (also called dipicolylamine; hereinafter “dpa”) for *Staphylococcus aureus* [19] or *Escherichia coli* [20,21] recognition. The dpa group, which is a phosphate sensor, forms a complex with phosphate and metal ions such as Cu^2+^ [22] or Zn^2+^ [23]. One research group has recently reported that some bacterial components were detected by dpa sensors using flow cytometry analysis [24]. The electrostatic interaction between the cationic amine group and the negatively charged bacterial surface has also been studied [25,26].

Saccharide recognition offers a potential solution because bacteria have specific glycolipids on their surface [27]. Phenylboronic acid, in particular, is known as a saccharide recognition site because it forms a bond with the *cis*-diol of saccharides [28,29]. In 2019, we reported that a chemically modified poly(amidoamine) (PAMAM) dendrimer having five phenylboronic acid groups at the terminus was able to recognize bacteria (Figure 1). The B-PAMAM dendrimer probe showed selectivity toward Gram-positive bacteria. The recognition was observed as the aggregate formation between bacteria and probe, which resulted in turbidity change [30]. Based on the results, we additionally introduced a fluorescent dansyl group on B-PAMAM and were able to achieve sensitivity improvement in fluorescence measurements [31]. However, the selectivity toward Gram-positive bacteria disappeared. We discovered that a PAMAM dendrimer without boronic acid formed minute aggregates with bacterial cells through electrostatic interaction. We therefore concluded that positively charged substituents, such as the terminal amine group of the PAMAM dendrimer or the dansyl group, interacted with the negatively charged bacterial surface. 

In order to achieve selective and sensitive recognition of Gram-positive bacteria, the effects of additional substituents introduced on B-PAMAM should be investigated. In this study, we introduced two substituents on B-PAMAM to examine nontargeted supramolecular interactions (betaine for electrostatic interaction and alkyl chain for hydrophobic interaction) (Figure 1) and clarified their effects on the selectivity toward bacteria under various pH conditions by turbidity measurement. To obtain higher selectivity and sensitivity, the dpa group was also introduced on B-PAMAM, which targeted phosphates on bacterial surface. We investigated the effects of nontargeted and targeted interactions on bacterial recognition by the ditopic PAMAM dendrimer probes.

## 2. Results and Discussion

### 2.1. Intermolecular Interactions between Dendrimer Probes and Bacteria

#### 2.1.1. Electrostatic Interaction

First, we investigated the electrostatic interaction between the dendrimer probes and the bacterial surface (Figure 2). As *S. aureus* [32,33] and *E. coli* [34,35] had negatively charged surfaces, a quaternary ammonium cation (betaine) group was introduced on B-PAMAM. The obtained product (Bt-B-PAMAM) was used for *S. aureus* and *E. coli* recognition at pH 3.0–10.5. Previous research has shown that *S. aureus* grew in a pH 4.0–9.8 environment [36] and *E. coli* grew in a pH 4.5–9.0 environment [37]. An *E. coli* strain was also found to tolerate acidic conditions (around pH 2–3) for a limited time [38]. Based on those studies, we decided not to examine strongly acidic or basic conditions because those conditions would be fatal to these bacteria.

The results indicated that B-PAMAM formed aggregates with *S. aureus* at pH 7–9. The binding force of boronic acid groups toward *S. aureus* was stronger than that toward *E. coli* and was responsible for the turbidity changes. Our results and explanations agreed with previous studies [30,31]. Compared with B-PAMAM, the cationic Bt-B-PAMAM showed low selectivity toward *S. aureus* under basic conditions. Bt-B-PAMAM formed aggregates not only with *S. aureus* but also with *E. coli* and caused turbidity changes at pH 8–10.5.

The boronic acid recognition sites for the two bacterial species would be different because Gram-positive bacteria and Gram-negative bacteria had different surface structures. The binding force of boronic acid groups was stronger for Gram-positive bacteria than toward Gram-negative bacteria. The differences in surface structures would result in the difference in the binding force of boronic acid groups. We noted that some boronic acid groups on PAMAM dendrimers formed boronate esters with saccharides on *E. coli* membrane surface, even under conditions that did not produce turbidity changes. The binding force for *E. coli* was not sufficient to form large aggregates or change the turbidity in the absence of electrostatic interaction. Although primary amine groups (p*K*_a_ = 8–9 [39]) on the dendrimer surface gradually lost their positive charges under basic conditions, the betaine groups of Bt-B-PAMAM were able to retain the electrostatic interaction with the bacterial surface. Therefore, the betaine groups would promote bacterial recognition regardless of pH. Because boronic acid groups easily formed boronate esters under basic conditions, Bt-B-PAMAM would produce turbidity changes when mixed with *E. coli* due to the strong binding force of the boronic acid groups and the electrostatic interaction of the betaine groups. The enhancement of bacterial recognition by the betaine groups seemed to be critical to exceed the threshold for turbidity change and resulted in a decrease of selectivity toward bacteria under basic conditions.

#### 2.1.2. Hydrophobic Interaction

We also investigated the hydrophobic interaction between the alkylated C-PAMAM and C-B-PAMAM probes and the bacterial surfaces (Figure 3). Because we expected that pH should not have a noticeable impact on the alkyl chain, we examined the conditions in which the pH range showed notable changes in Figure 2. 

We found that the bacterial suspensions containing C-PAMAM did not show a decrease in turbidity. This indicated that the hydrophobic interaction per se was not a critical driving force of the aggregation. In contrast, C-B-PAMAM formed aggregates with both *S. aureus* and *E. coli* in neutral (pH 7.4) and basic conditions. It was interesting that even though the alkyl undecane chains of the dendrimer probes were much longer than the boronic acid groups and were considered to cause steric hindrance, they did not disturb *S. aureus* and *E. coli* recognition. In the case of *E. coli*, the hydrophobic interaction between the alkyl chains of C-B-PAMAM and the *E. coli* bacterial surface had an impact on the recognition and the turbidity decrease. The outer membranes of Gram-negative bacteria contained hydrophobic lipopolysaccharide (LPS) that interacted with the alkyl chains. It was reported that the hydrophobic interaction with lipid backbone structures affected LPS recognition and detection sensitivity [25]. From our results, we conclude that the hydrophobic interaction per se did not cause aggregation and turbidity change; rather, it enhanced the recognition ability of boronic acid. The results of electrostatic interaction and hydrophobic interaction experiments demonstrated that nontargeted supramolecular interaction enhanced the recognition ability of boronic acid but had no notable impact on or resulted in the disappearance of selectivity toward bacteria. Hence, we set our sights on the design of ditopic dendrimer probes using dpa groups for phosphate-targeted recognition. 

### 2.2. dpa-Modified Ditopic Dendrimer Probes

#### 2.2.1. Bacterial Recognition by Cu-dpa-B-PAMAM

We investigated the recognition ability of a new ditopic B-PAMAM modified with dpa (dpa-B-PAMAM) (Figure 4). In order to improve sensitivity, the number of boronic acid modifications was increased compared with that of previous probes. Fifteen types of metal ions were added to dpa-PAMAM or dpa-B-PAMAM to form the corresponding chelates (M-dpa-PAMAM or M-dpa-B-PAMAM), and turbidity change before and after mixing was recorded.

Figure 4A shows that mixing M-dpa-PAMAM and *S. aureus* resulted in turbidity decrease, whereas mixing M-dpa-PAMAM and *E. coli* resulted in turbidity increase. The electrostatic interaction of the metal ions, the hydrophobic interaction of the dpa group, or the phosphate recognition would occur in M-dpa-PAMAMs. We considered that the formation of large aggregates that precipitated resulted in the turbidity decrease, and light scattering by the formed minute aggregates resulted in the turbidity increase (Appendix A). M-dpa modification was not superior to boronic acid modification because the turbidity reduction of the *S. aureus* suspension containing M-dpa-PAMAM did not reach a maximum. The results obtained using M-dpa-B-PAMAM, which contained both M-dpa and boronic acid groups, are shown in Figure 4B. All *S. aureus* suspensions containing M-dpa-B-PAMAM formed large aggregates, and turbidity was dramatically decreased. On the other hand, several M-dpa-B-PAMAMs, such as the Cu^2+^ chelate, remained dispersed in *E. coli* suspension but showed selectivity toward *S. aureus*. The results meant that some M-dpa-B-PAMAMs interacted with bacterial cells via phosphate recognition but not intermolecular interactions. Because the addition of Cu^2+^ ion to dpa-B-PAMAM resulted in good selectivity among the metal ions examined, Cu-dpa-B-PAMAM was considered the best ditopic dendrimer probe and used in further experiments.

We studied the recognition ability of Cu-dpa-B-PAMAM and found that it showed selectivity toward *S. aureus* over a wide pH range (Figure 5A). As demonstrated in Figure 5B, dpa-B-PAMAM did not show selectivity toward bacteria. From the results of C-B-PAMAM in Figure 3, we considered that the aggregation would be due to the hydrophobic interaction in cooperation with the recognition by boronic acid. In the case of dpa-B-PAMAM, the dpa group would behave as a hydrophobic substituent and contribute to the hydrophobic interaction. Compared with Cu-dpa-PAMAM (Figure 5C) and B-PAMAM (Figure 5D), Cu-dpa-B-PAMAM (Figure 5A and Appendix A) showed the highest selectivity over a wide pH range. That *E. coli* did not form aggregates with Cu-dpa-B-PAMAM under basic conditions suggested that Cu-dpa-B-PAMAM disrupted multipoint recognition and aggregation. We postulated that the probe was hidden or covered by the surface cell structure of *E. coli*. Further investigation is needed to elucidate the mechanism. Interestingly, from the standpoint of the number of boronic acid modifications, it is worth noting that B-PAMAM in Figure 5D showed loss of selectivity toward bacteria under basic conditions, in contrast to B-PAMAM having a small number of boronic acid modifications in Figure 2. The larger number of modifications seemed to result in stronger recognition ability, which led to the formation of large aggregates with both *S. aureus* and *E. coli*. As phenylboronic acid’s p*K*_a_ is approximately 8, pH conditions also enhanced the recognition ability. Hence, even though *E. coli* struggled to form aggregation, the two factors mentioned above produced turbidity changes in Figure 5D. The results demonstrated that the ditopic strategy, that is, cooperation between the Cu-dpa group and the boronic acid group, was important to obtain selectivity toward bacteria.

Then, we investigated the LOD using turbidity measurements (Appendix A) and microscopic observations (Appendix A). Turbidity changes were observed in the 2.3 × 10^6^ CFU·mL^−1^
*S. aureus* suspension containing Cu-dpa-B-PAMAM, as indicated in Appendix A. Microscopy images of suspensions having low concentrations of *S. aureus* also showed minute aggregates (Appendix A), whereas *E. coli* suspensions did not demonstrate any changes (Appendix A). Hence, the results suggested that the detection limit of *S. aureus* was 2.3 × 10^6^ CFU·mL^−1^ by turbidity measurement, but the aggregate formation could occur at a lower concentration, such as 2.3 × 10^5^ CFU·mL^−1^. Moreover, Cu-dpa-B-PAMAM retained its selectivity toward bacteria, which was derived from the difference in recognition ability for different bacterial surfaces. As the detection limit of bacteria using the monotopic B-PAMAM dendrimer probe was 6.4 × 10^6^ CFU·mL^−1^ [30], Cu-dpa-B-PAMAM exhibited improved sensitivity over a wide pH range while retaining its selectivity toward *S. aureus*.

#### 2.2.2. Selectivity toward Gram-Positive Bacteria and Assay for Viability

We investigated the selectivity toward bacteria by using several bacterial species. The results of turbidity measurements and the images of aggregates are shown in Figure 6 and Appendix A, respectively. Turbidity was decreased in all Gram-positive bacterial suspensions but not in Gram-negative ones. We considered that the turbidity increase in Gram-negative bacterial suspensions was caused by the formation of minute aggregates. Therefore, the bacterial recognition ability was not specific for *S. aureus* IAM1011 or *E. coli* K12W3110 used in the initial experiments. The turbidity measurements confirmed that Cu-dpa-B-PAMAM was able to selectively discriminate Gram-positive bacteria from Gram-positive bacteria.

We conducted an MTT assay to determine cell viability after the recognition procedure (Figure 7). MTT added to bacterial suspension would be converted into formazan by live cells in the suspension. As formazan produced maximum absorbance at 560 nm, an absorbance decrease at this wavelength would denote cell death. In all experiments, the viability of *S. aureus* was higher than that of *E. coli*, and there was no correlation between cell viability and turbidity change. The results were consistent with a previous report [31] indicating that *E. coli* was more sensitive to environmental stress than *S. aureus*. Our recognition method using boronic acid-modified PAMAM dendrimers affected the viability of *S. aureus* but there was no relationship between bacteria viability and the size of aggregation. Our method is convenient because it is able to selectively recognize and collect live Gram-positive bacteria cells in 20 min. Our method enables rapid detection and would be useful in the initial screening for bacterial species and strains. 

## 3. Materials and Methods

### 3.1. Reagents and Apparatus

#### 3.1.1. Reagents

All reagents and solvents were purchased from commercial suppliers and used without further purification unless otherwise specified. Betaine hydrochloride (023-01712), 4-carboxyphenylboronic acid (354-36403), sodium chloride (191-01665), and agar (018-15811) were purchased from Fujifilm Wako Pure Chemical Corporation (Osaka, Japan). N-cyclohexyl-3-aminopropanesulfonic acid (CAPS, GB06), 4-(2-hydroxyethyl)-1-piperazineethanesulfonic acid (HEPES, GB10), dimethylsulfoxide luminasol (DMSO-Lu, LU08), 3-(4,5-dimethyl-2-thiazolyl)-2,5-diphenyl-2H-tetrazolium bromide (MTT, M009), and 4′,6-diamidino-2-phenylindole dihydrochloride n-hydrate (DAPI, D523) were purchased from Dojindo Laboratories (Kumamoto, Japan). 4-Hydroxybenzoic acid (H0207), 2,2′-dipicolylamine (D2228), formaldehyde solution (37%, F0622), undecanoic acid (U0004), and 4-(4,6-dimethoxy-1,3,5-triazin-2-yl)-4-methylmorpholinium chloride (DMT-MM, D2919) were purchased from Tokyo Chemical Industry Co., Ltd. (Tokyo, Japan). Acetonitrile (01837-25), chloroform-d_1_ (07660-23), methanol (25183-70), methanol-*d*_4_ (99.8%, 25183-70), and D_2_O (99.8%, 10086-23) were purchased from Kanto Chemical Co. Inc. (Tokyo, Japan). Poly(amidoamine) (PAMAM) dendrimer, ethylenediamine core, and generation 4.0 (412449-10G) solution was purchased from Sigma-Aldrich Japan Co. LLC. (Tokyo, Japan). Bacto yeast extract (212750) and Bacto tryptone (211705) were purchased from Nippon Becton, Dickinson Co., Ltd. (Tokyo, Japan). Phosphate-buffered saline (PBS, 2101) was purchased from Cell Science & Technology Institute, Inc. (Miyagi, Japan). Water was doubly distilled and deionized using a Milli-Q water system (WG222, Yamato Scientific Co., Ltd., Tokyo, Japan and Milli-Q Advantage, Merck Millipore, MA, USA) before use. Spectra/Por 6 dialysis bag (MW cutoff = 1000) was purchased from Repligen Co. (Rancho Dominguez, CA, USA).

#### 3.1.2. Apparatus

^1^H nuclear magnetic resonance (NMR) spectra were recorded on a JEOL JNM-ECA 500 spectrometer (500 MHz; JEOL, Tokyo, Japan) at 300 K or a Bruker Avance III HD 400 MHz at 298 K in corresponding deuterated solvents. All pH values were recorded using a Horiba F-52 pH meter (HORIBA Ltd., Kyoto, Japan). Ultraviolet–visible (UV–Vis) absorption spectra were measured using a JASCO V-570 or V-760 UV–Vis spectrophotometer (JASCO Co., Tokyo, Japan) equipped with a Peltier thermocontroller and a 10 mm quartz cell. Samples were automatically mixed in an MS-300 Multi Shaker (AS ONE Co., Osaka, Japan). Centrifugation was conducted using CF15RN (Hitachi High-Technologies Co., Tokyo, Japan). Nanoprobes were lyophilized by EYELA FDU-1200 (Tokyo Rikakikai Co., Ltd., Tokyo, Japan). Fluorescence microscopy images were obtained using an Axiovert 200 (Carl Zeiss Co., Ltd., Oberkochen, Germany).

### 3.2. Chemical Syntheses

#### 3.2.1. Synthesis of B-PAMAMs

4-Carboxyphenylboronic acid and DMT-MM were dissolved in methanol (10 mL), and the reaction mixture was stirred at room temperature for 1 h [30,31]. PAMAM(G4) dendrimer (1 mL, 1 eq, 5.73 μmol) was added, and the reaction mixture was stirred at room temperature for 2 days (Table 1). The reaction mixture was transferred into a Spectra/Por 6 dialysis bag and dialyzed against methanol and distilled water for several days. The purified product was lyophilized to give white flocks, the chemical structure of which was confirmed by ^1^H NMR measurement. The number of phenylboronic acid substituents was calculated from the peak area of the ^1^H NMR spectra (Appendix A). 

#### 3.2.2. Synthesis of Bt-B-PAMAM

Betaine hydrochloride (26.7 mg, 123 eq, 174 μmol) and DMT-MM (74.0 mg, 189 eq, 267 μmol) were dissolved in methanol (5 mL), and the reaction mixture was stirred at room temperature for 30 min. B4-PAMAM (20.5 mg, 1.0 eq, 1.41 μmol) dissolved in methanol (5 mL) was added, and the reaction mixture was stirred at room temperature for 2 days. The reaction mixture was transferred into a Spectra/Por 6 dialysis bag and dialyzed against methanol and distilled water for several days. The purified product was lyophilized to give white flocks (19.3 mg), the chemical structure of which was confirmed by ^1^H NMR measurement. The number of betaine substituents was calculated to be 33 from the peak area of the ^1^H NMR spectrum (Appendix A).

#### 3.2.3. Synthesis of C-PAMAM

Undecanoic acid (21.0 mg, 10 eq, 112.7 μmol) and DMT-MM (123.0 mg, 40 eq, 444.5 μmol) were dissolved in methanol (9 mL), and the reaction mixture was stirred at room temperature for 30 min. PAMAM(G4) dendrimer (2 mL, 1 eq, 11.5 μmol) was added, and the reaction mixture was stirred at room temperature for 2 days. The reaction mixture was transferred into a Spectra/Por 6 dialysis bag and dialyzed against methanol and distilled water for several days. The purified product was lyophilized to give white flocks (208.4 mg), the chemical structure of which was confirmed by ^1^H NMR measurement. The number of substituents was calculated to be 10 from the peak area of the ^1^H NMR spectrum (Appendix A).

#### 3.2.4. Synthesis of C-B-PAMAM

4-Carboxyphenylboronic acid (8.0 mg, 8 eq, 48.2 μmol) and DMT-MM (64.0 mg, 40 eq, 231.2 μmol) were dissolved in methanol (5 mL), and the reaction mixture was stirred at room temperature for 30 min. C-PAMAM (106.6 mg, 1.0 eq) was added, and the reaction mixture was stirred at room temperature for 3 days. The reaction mixture was transferred into a Spectra/Por 6 dialysis bag and dialyzed against methanol and distilled water for several days. The purified product was lyophilized to give white flocks (102.9 mg), the chemical structure of which was confirmed by ^1^H NMR measurement. The number of boronic acid substituents was calculated to be 6 from the peak area of the ^1^H NMR spectrum (Appendix A). 

#### 3.2.5. Synthesis of dpa

2,2′-Dipicolylamine (804 mg, 1 eq, 4.04 mmol) was dissolved in 20 mL of acetonitrile, and the reaction mixture was stirred at 100 °C for 2 h. Formaldehyde (368 mg, 1.1 eq, 4.53 mmol) was added, and the reaction mixture was stirred for another 1.5 h. The reaction mixture was cooled to 55 °C, and 4-hydroxyphenylboronic acid (864 mg, 1.5 eq, 6.25 mmol) dissolved in acetonitrile (30 mL) was added. The reaction mixture was stirred at 55 °C for 3 days. This was followed by concentration in vacuo and extraction with chloroform four times. The organic layer was concentrated in vacuo to obtain a yellow solid (1.13 g, 87%, 3.23 mmol). The chemical structure was confirmed from the ^1^H NMR spectrum (Appendix A), and ESI-HRMS (*m*/*z*) calcd for C_20_H_19_N_3_O_3_Na[M+Na]^+^ 372.1324, found 372.1303 (Appendix A).

#### 3.2.6. Synthesis of dpa-PAMAM

Synthetic dpa (37.4 mg, 9.4 eq, 107 μmol) and DMT-MM (65.5 mg, 20.7 eq, 237 μmol) were dissolved in methanol (10 mL), and the reaction mixture was stirred at room temperature for 1 h. PAMAM(G4) dendrimer (2 mL, 1 eq, 11.5 μmol) dissolved in methanol (5 mL) was added, and the reaction mixture was stirred at room temperature for 2−3 days. The reaction mixture was transferred into a Spectra/Por 6 dialysis bag and dialyzed against methanol and distilled water for several days. The purified product was lyophilized to give white flocks (77.8 mg), the chemical structure of which was confirmed by ^1^H NMR measurement. The number of dpa substituents was calculated to be four from the peak area of the ^1^H NMR spectrum (Appendix A).

#### 3.2.7. Synthesis of dpa-B-PAMAM

Synthetic dpa (5.42 mg, 9.83 eq, 32.4 μmol) and DMT-MM (45.0 mg, 48.9 eq, 162 μmol) were dissolved in methanol (10 mL), and the reaction mixture was stirred at room temperature for 1 h. B7-PAMAM (51.3 mg, 1.0 eq, 3.32 μmol) dissolved in methanol (5 mL) was added, and the reaction mixture was stirred at room temperature for 2 days. The reaction mixture was transferred into a Spectra/Por 6 dialysis bag and dialyzed against methanol and distilled water for several days. The purified product was lyophilized to give white flocks (52.1 mg), the chemical structure of which was confirmed by ^1^H NMR measurement. The number of dpa substituents was calculated to be four from the peak area of the ^1^H NMR spectrum (Appendix A).

### 3.3. Biological Experiments

#### 3.3.1. Cell Culture

Lysogeny broth (LB) was a mixture of 2 g of Bacto tryptone, 1 g of Bacto yeast extract, and 2 g of NaCl in 200 mL of distilled water. *S. aureus* IAM1011, *S. aureus* ATCC25923, *S. aureus* ATCC29213, *S. pseudintermedius* 2012-S-27, *S. epidermidis* ATCC12228, *Enterococcus faecalis* ATCC29212, *E. coli* K12W3110, *E. coli* ATCC25922, *Pseudomonas aeruginosa* ATCC27853, and *Salmonella enteritidis* ATCC13311 were provided by RIKEN BRC (Ibaraki, Japan) and cultured at 37 °C overnight on LB agar plates. For each experiment, we used *S. aureus* IAM1011 as *S. aureus* and *E. coli* K12W3110 as *E. coli* unless otherwise specified. A mixture of 200 mL of LB and 3 g of agar was used to prepare an LB plate. Cultured colonies were picked up and isolated followed by an overnight culture in LB at 37 °C. The suspension was centrifuged (10,000 rpm, 1 min) and washed with distilled water. This washing protocol was repeated twice. The corresponding buffer was added to the washed cells and the bacterial suspension was centrifuged. The concentration of the bacterial suspension was adjusted by measuring OD_600_ after vortex mixing. The calculation index is shown below.

*S. aureus*: OD_600_ = 1.0, CFU = 4.5 × 10^8^ mL^−1^.*E. coli*: OD_600_ = 1.0, CFU = 1.0 × 10^9^ mL^−1^.

The cultured cells were used in the following experiments.

#### 3.3.2. Bacterial Detection

PAMAM dendrimer probes (750 mL, 6.6 μM) and *S. aureus* or *E. coli* in buffer solution (750 μL, 2.3 × 10^5^ to 2.3 × 10^8^ CFU·mL^−1^) were mixed and used as reference samples for turbidity measurement [30,31]. It should be noted that C-PAMAM or C-B-PAMAM sample solution produced lather during sample preparation. Hence, the reference samples for these two dendrimer probes were prepared using 750 μL of corresponding bacterial suspension and 750 μL of PBS buffer. All dendrimer reference samples except C-PAMAM or C-B-PAMAM reference samples were mixed at 2000 rpm for 10 min. Turbidity measurements or fluorescence microscopy observation was conducted after standing for 10 min. For fluorescence microscopy, DAPI solution was mixed with cultured cells in PBS buffer, and excess DAPI was washed off before the observation. ΔOD_600_ was expressed as the difference in optical density of a sample before (or reference samples in C-PAMAM and C-B-PAMAM) and after mixing. 

ΔOD_600_ = OD_600_(after) − OD_600_(before or control).Turbidity change = ΔOD_600_/OD_600_(before or control).

#### 3.3.3. MTT Assay 

Bacterial cells (750 μL, 2.0 × 10^8^ CFU·mL^−1^ for *S. aureus* and 4.5 × 10^8^ CFU·mL^−1^ for *E. coli*) and dendrimer probe solution (750 μL, 6.6 μM) or HEPES buffer (750 μL) were mixed for 10 min at 2000 rpm to form aggregates [31]. The suspension was centrifuged to separate the supernatant from the residue. LB (750 μL). MTT solution (100 μL, 0.4 mg·mL^−1^) was added to the residue, and the suspension was incubated at 37 °C for 10 min (*S. aureus*) or 15 min (*E. coli*). After the incubation, the suspension was centrifuged, and the supernatant was separated from the residue. As the final step, 200 μL of DMSO(Lu) was added to destroy the cell membrane and quench the reaction. The absorbance of dissolved formazan was measured (*λ*_abs_ = 560 nm), and the absorbance of the control samples was set at 100%.

#### 3.3.4. Statistical Analyses of Data

Each statistical analysis was conducted with an alpha level of 0.05 using R version 3.6.2 [40]. The differences in turbidity measurement between Gram-positive and Gram-negative bacteria suspensions in Figure 6 were analyzed by two-sided Welch’s *t*-test. The differences in bacteria viability in Figure 7 were analyzed by two-way analysis of variance (ANOVA) and Tukey’s HSD test after Bartlett’s test.

## 4. Conclusions

We have developed a rapid and selective bacterial recognition method for Gram-positive bacteria that uses the ditopic Cu-dpa-B-PAMAM dendrimer probe. The method was able to discriminate Gram-positive bacteria from Gram-negative bacteria. The method was simple as it required only mixing and standing, and the results could be obtained within 20 min. Furthermore, the Cu-dpa-B-PAMAM dendrimer probe had higher sensitivity and could be utilized over a wider pH range than monotopic B-PAMAM probe or Cu-dpa-PAMAM dendrimer probe. The applicable pH range of the Cu-dpa-B-PAMAM dendrimer probe almost covered the pH range for bacterial viability.

We also investigated the supramolecular interaction of B-PAMAM and found that neither electrostatic interaction nor hydrophobic interaction was the main driving force for the aggregation and the turbidity changes. These two interactions did not affect turbidity alone but cooperated with boronic acid to enhance the recognition ability of B-PAMAM. The enhancement of nontargeted interactions did not improve selectivity toward bacteria, whereas the addition of the Cu-dpa moiety that selectively recognized phosphates resulted in improved selectivity toward *S. aureus*. Therefore, we conclude that a targeted design, such as Cu-dpa-B-PAMAM, would improve selective and sensitive recognition for Gram-positive bacteria by the ditopic B-PAMAM dendrimer probe.

We conducted bacteria viability measurements and found that the probes affected the viability of *E. coli* in particular. *E. coli* was more sensitive to environmental stress than *S. aureus*. The method developed herein would enable rapid and species-specific detection of bacteria and be useful in the initial screening for bacterial species and strains. Because our method could be utilized over a wide pH range, it would be a better choice than conventional methods that use biological reagents such as antibodies or specific proteins. We also expect various applications including testing for infection using urine samples and food sanitation experiments.

## Figures and Tables

**Figure 1 molecules-27-00256-f001:**
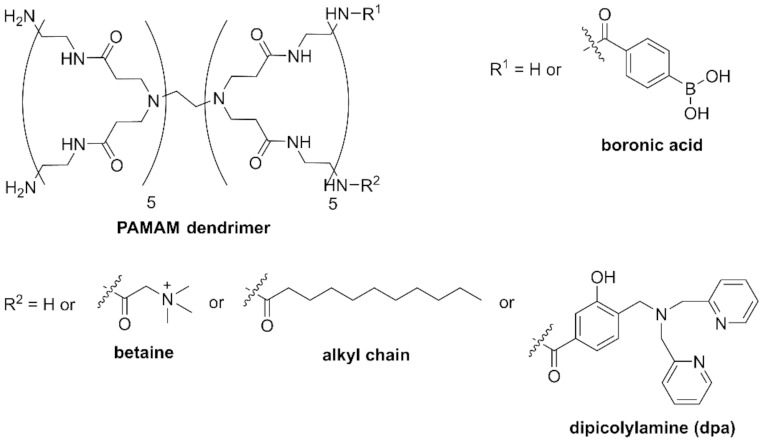
Structures of PAMAM dendrimer probes.

**Figure 2 molecules-27-00256-f002:**
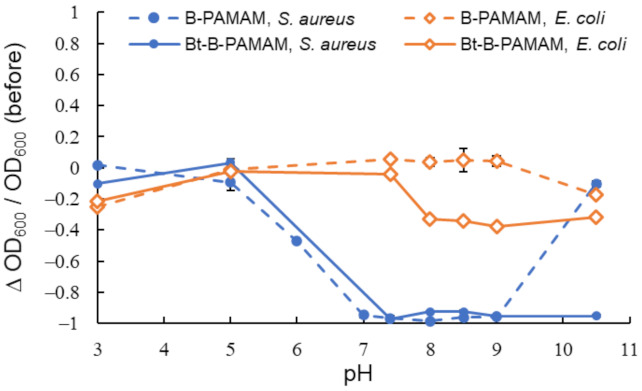
Turbidity changes in bacterial suspensions containing B-PAMAM and Bt-B-PAMAM in PBS buffer (*n* = 6). [probe] = 3.3 μM; [bacteria] = 2.3 × 10^8^ CFU·mL^−1^. Bars are expressed as standard deviation (SD) for six separate experiments. The number of betaine modifications was 33 and that of boronic acid modifications was 4.

**Figure 3 molecules-27-00256-f003:**
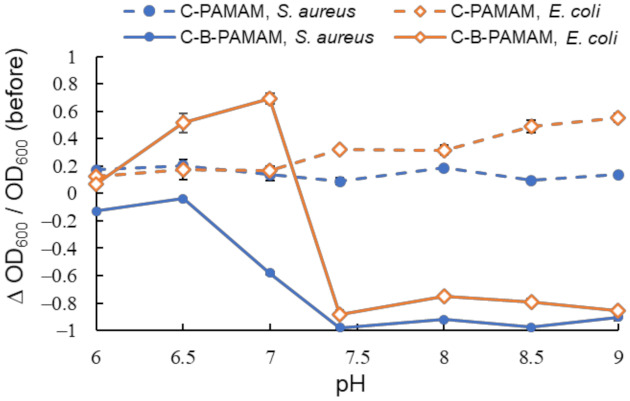
Turbidity changes in bacterial suspensions containing C-PAMAM and C-B-PAMAM in PBS buffer (*n* = 3). [probe] = 3.3 μM; [bacteria] = 2.3 × 10^8^ CFU·mL^−1^. Bars are expressed as SD for three separate experiments. The number of alkyl chain modifications was 10 and that of boronic acid modifications was 6.

**Figure 4 molecules-27-00256-f004:**
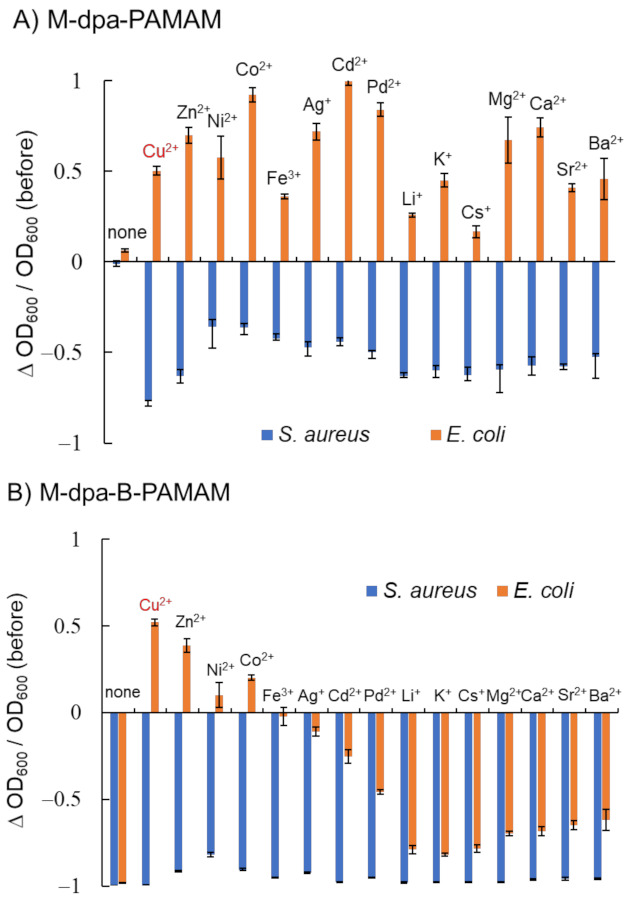
Turbidity changes in bacterial suspensions containing dpa-modified probes at pH 7.4 adjusted with HEPES buffer (*n* = 3). [probe] = 3.3 μM; [bacteria] = 2.3 × 10^8^ CFU·mL^−1^; [(M)(NO_3_)*_n_*] = none or 13.2 μM; [HEPES] = 5.0 mM. Bars are expressed as SD for three separate experiments. (**A**) Effects of metal ions on dpa-PAMAM. (**B**) Effects of metal ions on dpa-B-PAMAM. The number of dpa modifications was 4 and that of boronic acid modifications was 7.

**Figure 5 molecules-27-00256-f005:**
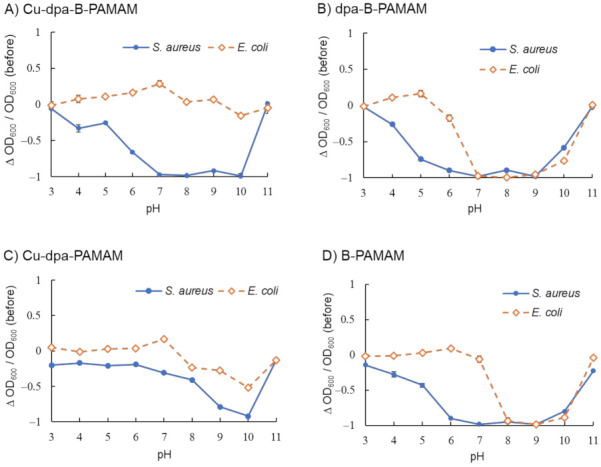
Turbidity changes in bacterial suspensions containing Cu-dpa-B-PAMAM (*n* = 3). [probe] = 3.3 μM; [Cu^2+^] = 13.2 μM or none; [bacteria] = 2.3 × 10^8^ CFU·mL^−1^; [HEPES or CAPS] = 5.0 mM. Bars are expressed as SD for three separate experiments. (**A**) Cu-dpa-B-PAMAM, (**B**) dpa-B-PAMAM, (**C**) Cu-dpa-PAMAM, (**D**) B-PAMAM. The number of dpa modifications was 4 and the number of boronic acid modifications was 7 for (**A**−**C**) and 8 for (**D**).

**Figure 6 molecules-27-00256-f006:**
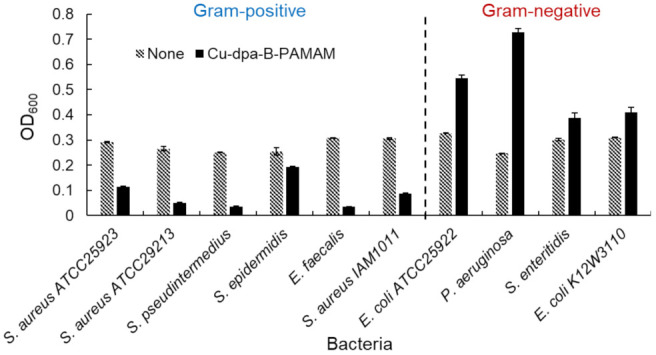
Turbidity experiments using various bacteria (*n* = 3). [Cu-dpa-B-PAMAM] = none or 3.3 μM; [Cu^2+^] = none or 13.2 μM; bacterial concentration was set at OD_600_ = 0.3; [HEPES] = 5.0 mM. Bars are expressed as SD for three separate experiments. Each bacterial suspension containing Cu-dpa-B-PAMAM could be discriminated from the suspension without the probe (*p* < 0.05: *S. aureus* ATCC25923: 1.2 × 10^−7^, *S. aureus* ATCC29213: 4.5 × 10^−4^, *S. pseudintermedius*: 1.0 × 10^−8^, *S. epidermidis*: 1.4 × 10^−4^, *E. faecalis*: 1.4 × 10^−9^, *S. aureus* IAM1011: 7.3 × 10^−5^, *E. coli* ATCC25922: 9.6 × 10^−4^, *P. aeruginosa*: 5.6 × 10^−4^, *S. enteritidis*: 2.6 × 10^−2^, *E. coli* K12W3110: 1.5 × 10^−^^2^). The suspensions containing Gram-positive bacteria and Cu-dpa-B-PAMAM could be discriminated from suspensions containing Gram-negative bacteria and Cu-dpa-B-PAMAM (*p* < 0.05). The number of dpa modifications was 4 and that of boronic acid modifications was 7.

**Figure 7 molecules-27-00256-f007:**
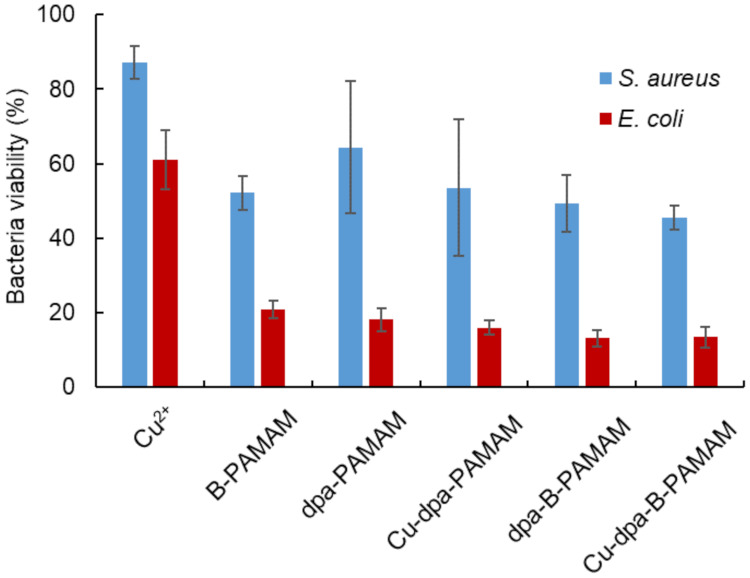
Bacteria viability at pH 7.4 measured by MTT assay (*λ*_abs_ = 560 nm, *n* = 3). [probe] = 3.3 μM or none; [Cu^2+^] = 13.2 μM or none; [*S. aureus*] = 5.0 × 10^7^ CFU·mL^−1^; [*E. coli*] = 2.3 × 10^8^ CFU·mL^−1^; [HEPES] = 5.0 mM. Bars are expressed as SD for three separate experiments. The viability of *S. aureus* could be discriminated from the results in *E. coli* (*p* < 0.05). The viability of suspensions contains Cu^2+^ could be discriminated from the suspensions containing other probes (*p* < 0.05).

**Table 1 molecules-27-00256-t001:** Syntheses of B-PAMAM dendrimers.

Probe	PAMAM Dendrimer	Phenylboronic Acid	DMT-MM	Number of Modifications	Yield
B4-PAMAM	1 mL (1 eq, 5.73 μmol)	5.72 mg (6 eq, 34.5 μmol)	48.0 mg (30 eq, 174 μmol)	4	73.6 mg
B7-PAMAM	2 mL (1 eq, 11.46 μmol)	15.18 mg (8.2 eq, 93.3 μmol)	227.7 mg (72 eq, 823 μmol)	7	101 mg
B8-PAMAM	2 mL (1 eq, 11.46 μmol)	18.97 mg (10 eq, 114 μmol)	158 mg (50 eq, 572 μmol)	8	126 mg

## Data Availability

Data sharing not applicable.

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
