# Peer review of "Rapid Bacterial Recognition over a Wide pH Range by Boronic Acid-Based Ditopic Dendrimer Probes for Gram-Positive Bacteria"

_molecules, 2021, doi:10.3390/molecules27010256_

Round 1

Reviewer 1 Report

The manuscript by Mikagi et al. presents a study of the possibility of using modified ditopic poly (amidoamine) (PAMAM) for selective recognition of bacteria. I find the presented work very interesting and worthy of publication in Molecules journal.
However, I would like to clarify a few questions:
Firstly, Figure 2 clearly demonstrates that phenylboronic acid-modified PAMAM (B-PAMAM) does not cause changes in the turbidity of e-coli solutions. At the same time, in a similar experiment in Figure 5, significant changes are noticeable at PH 8-10. Please clarify this.
The authors also point out that alkylated B-PAMAM (C-B-PAMAM) interacts with e-coli, while alkylated PAMAM (C-PAMAM), like B-PAMAM, does not. Can the authors suggest a possible mechanism for this effect?

Author Response

We deeply appreciate to you for the insightful comments on our paper. The followings are responses to the comments:

1) Firstly, Figure 2 clearly demonstrates that phenylboronic acid-modified PAMAM (B-PAMAM) does not cause changes in the turbidity of e-coli solutions. At the same time, in a similar experiment in Figure 5, significant changes are noticeable at PH 8-10. Please clarify this.

We thank a lot for your kind comment. We consider that two factors resulted in the turbidity changes in E. coli suspension of Figure 5 (but no change in Figure 2).

First, the number of phenylboronic acid modifications of B-PAMAM was different between Figure 2 (page 3, line 100) and Figure 5 (page 6, line 211). Since B-PAMAM in Figure 5 had more phenylboronic acids, it had stronger recognition ability than B-PAMAM in Figure 2. As the second factor, phenylboronic acid (pKa is approximately 8) could easily form boronate ester with bacteria’s saccharides at pH 8-10 (page 3, line 120-121). Hence, even though E. coli was difficult to form aggregation, the two factors gave turbidity changes in Figure 5.

The explanation was also added to the main text.

Page 6, line 200-204: “The larger number of modifications seemed to result in stronger recognition ability, which led to the formation of large aggregates with both S. aureus and E. coli. Since phenylboronic acid’s pKa is approximately 8, pH conditions also enhanced the recognition ability. Hence, even though E. coli was difficult to form aggregation, the two factors mentioned above produced turbidity changes in Figure 5D.”

2) The authors also point out that alkylated B-PAMAM (C-B-PAMAM) interacts with e-coli, while alkylated PAMAM (C-PAMAM), like B-PAMAM, does not. Can the authors suggest a possible mechanism for this effect?

We appreciate your interest in this point. We consider that phenylboronic acid of B-PAMAM had the potential to recognize E. coli (Figure 5D, basic conditions) but it was not sufficient alone at neutral or acidic conditions (Figure 2 and 5D). Our results and ref 25 (Liu, L. et al., Nat. Nanotechnol., 2009) suggested that C-B-PAMAM’s undecane chain could interact with hydrophobic LPS and it assisted C-B-PAMAM contact and recognize E. coli. Therefore, even though B-PAMAM could not recognize E. coli, C-B-PAMAM recognized E. coli and gave turbidity changes (Figure 3). In the case of C-PAMAM, it has only hydrophobic interaction and could not form a covalent bond, resulting in no aggregation and turbidity changes (Figure 3). Please also see the discussion in page 4, line 142-152.

Reviewer 2 Report

In this paper, the authors developed a dendrimer probe for rapidly distinguishing Gram-positive bacteria from Gram-negative bacteria with a wide pH range. Although it does not show good selectivity on S. aureus as expected, it could be used for detecting Gram-positive bacteria. In addition, the authors also investigated the supramolecular interaction between probe and bacteria, which can be useful for improving the selective and sensitive of new probes for bacteria.

However, there is still some problem to be addressed before the publication in Molecules.

  1. Title: If possible, I suggested a title of “Rapid Bacterial Recognition over a Wide pH Range by Boronic Acid-Based Ditopic Dendrimer Probes for Gram-positive bacteria” because the probe dose not showed a specificity on one subtype of Gram-positive or Gram-negative bacteria.
  2. Introduction: Page 2, line 60. Please cite the reference here.
  3. Results and Discussion: Page 4, line 138, can you give some explanation why the long alkyl undecane chain did not disturb S. aureus and E. coli recognition.
  4. Figure 6. Please add the P value summary on different pairs of columns.
  5. Figure 7. The author described their probe did not adversely affect the viability of S. aureus, however, the bacteria viability was obviously decreased as shown in Fig. 7. If possible, the author should rewrite this sentence.
  6. Method: Can author provided the HRMS of their compounds?
  7. Conclusion: Page 6, line 417, as mentioned in Q1, the word “selective” was insufficient.

Author Response

We deeply appreciate your insightful comments on our paper. The comments have helped us significantly improve the paper. In particular, we wish to acknowledge your highly valuable comments for clarification on the meaning of “selective” toward bacteria. The followings are responses to the comments:

1) Title: If possible, I suggested a title of “Rapid Bacterial Recognition over a Wide pH Range by Boronic Acid-Based Ditopic Dendrimer Probes for Gram-positive bacteria” because the probe dose not showed a specificity on one subtype of Gram-positive or Gram-negative bacteria.

We thank you for the kind comment, and we changed the title to the suggested “Rapid Bacterial Recognition over a Wide pH Range by Boronic Acid-Based Ditopic Dendrimer Probes for Gram-Positive Bacteria” to avoid readers’ misunderstanding.

2) Introduction: Page 2, line 60. Please cite the reference here.

According to the comment, a reference book was added to page 2, line 61 as ref 27 (Dumitriu, S. Eds.; Polysaccharides: Structural Diversity and Functional Versatility, Second Edition.; CRC Press: United States, 2014, Chapter 1.).

3) Results and Discussion: Page 4, line 138, can you give some explanation why the long alkyl undecane chain did not disturb S. aureus and E. coli recognition.

We appreciate your concerns on this point. Even though further investigation is still needed, our results (Figure 3 and 5B) and ref 25 (Liu, L. et al., Nat. Nanotechnol., 2009) suggested that the undecane chain interacted with bacterial hydrophobic structure. So far, we consider that hydrophobic interaction might fix the conformation of the undecane chain, and it resulted in reducing the chain’s steric hindrance.

4) Figure 6. Please add the P value summary on different pairs of columns.

In accordance with the comment, P-value of each bacterium was added to the caption of Figure 6 as below. The suspensions containing Cu-dpa-B-PAMAM and bacteria were not compared with each other by separate t-tests to avoid the risk of Type 1 error.

Page 7, line 237-240: “S. aureus ATCC25923: 1.2 × 10-7, S. aureus ATCC29213: 4.5 × 10-4, S. pseudintermedius: 1.0 × 10-8, S. epidermidis: 1.4 × 10-4, E. faecalis: 1.4 × 10-9, S. aureus IAM1011: 7.3 × 10-5, E. coli ATCC25922: 9.6 × 10-4, P. aeruginosa: 5.6 × 10-4, S. enteritidis: 2.6 × 10-2, E. coli K12W3110: 1.5 × 10-2

5) Figure 7. The author described their probe did not adversely affect the viability of S. aureus, however, the bacteria viability was obviously decreased as shown in Fig. 7. If possible, the author should rewrite this sentence.

According to the comment, we rewrote the sentence as below.

Page 8, line 250-253: “Our recognition method using boronic acid-modified PAMAM dendrimers affected the viability of S. aureus but there was no relationship between bacteria viability and the size of aggregation.”

6) Method: Can author provided the HRMS of their compounds?

As requested, we newly provided a ESI-HRMS result of dpa in the main text and Figure S14.

Page 10, line 350-351: “m/z calcd for C20H19N3O3Na[M+Na]+ 372.1324, found 372.1303 (Figure S14).”

Regrettably, however, since other synthetic compounds were large molecules (m/z > 10000), we could not get MS spectra by our ESI-MS or FAB-MS instruments.

7) Conclusion: Page 6, line 417, as mentioned in Q1, the word “selective” was insufficient.

We appreciate the suggestive comment, and we rewrote “selective” into “selective for Gram-positive bacteria” to clarify the meaning as below.

Page 12, line 423-424: “We have developed a rapid and selective bacterial recognition method for Gram-positive bacteria that uses the ditopic Cu-dpa-B-PAMAM dendrimer probe.”

Page 12, line 437-439: “Therefore, we conclude that a targeted design such as Cu-dpa-B-PAMAM would improve selective and sensitive recognition for Gram-positive bacteria by the ditopic B-PAMAM dendrimer probe.”

Round 2

Reviewer 1 Report

The authors answered all my questions, so I believe that this manuscript can be published in the presented form.